# A Concise Diastereoselective Total Synthesis of α-Ambrinol

**DOI:** 10.3390/md21040230

**Published:** 2023-04-01

**Authors:** Josefa L. López-Martínez, Irene Torres-García, Irene Moreno-Gutiérrez, Pascual Oña-Burgos, Antonio Rosales Martínez, Manuel Muñoz-Dorado, Míriam Álvarez-Corral, Ignacio Rodríguez-García

**Affiliations:** 1Organic Chemistry, University of Almería, CIAIMBITAL, 04120 Almería, Spain; pepaloma91@hotmail.com (J.L.L.-M.); irene.tg.94@gmail.com (I.T.-G.); irenemorenogtrz@gmail.com (I.M.-G.); mdorado@ual.es (M.M.-D.); 2Instituto de Tecnología Química, Universitat Politècnica de València-Consejo Superior de Investigaciones Científicas (UPV-CSIC), 46022 Valencia, Spain; pasoabur@itq.upv.es; 3Department of Chemical Engineering, Escuela Politécnica Superior, University of Sevilla, 41011 Sevilla, Spain; arosales@us.es

**Keywords:** ambergris, ambrinol, barbier reaction, CpTiCl_2_

## Abstract

(−)-*cis*-α-Ambrinol is a natural product present in ambergris, a substance of marine origin that has been highly valued by perfumers. In this paper, we present a new approach to its total synthesis. The starting material is commercially available α-ionone and the key step is an intramolecular Barbier-type cyclization induced by CpTiCl_2_, an organometallic compound prepared in situ by a CpTiCl_3_ reduction with Mn.

## 1. Introduction

Ambergris is a solid that forms pathogenically in the digestive tract of sperm whales (*Physeter macrocephalus*). Although it has an unpleasant odor when fresh, as it dries and ages it emits a pleasant and subtle fragrance, which is the reason why it is highly prized in the perfume industry. On average, ambergris is present in one out of every hundred sperm whales of both sexes. It has been documented that of the 1933 sperm whales captured between 1934 and 1953, only 19 had ambergris, with a total weight of approximately 1155 kg [1]. Due to the prized properties of ambergris as a fixative and stabilizer in perfumery, its chemical composition has been the subject of numerous studies. Although in variable proportions, the main constituents include a group of steroids with a cholestane skeleton, and the squalene-derived triterpene ambrein (**2**). In fact, although ambrein (**2**) has been prepared in vitro by enzymatic cyclization of squalene (**1**) (Figure 1a), ^13^C isotopic ratio studies suggest that its biosynthesis [2], unlike the co-occurring steroids, takes place with the participation of some bacteria and it is not exclusive to the sperm whale, a result which is in line with considering ambergris as a product resulting from a pathology of the marine mammal [3].

Although ambrein (**2**) is an odorless compound, the aroma of ambergris is due to products formed through a natural oxidative degradation, possibly as a result of its exposure to air, seawater and sunlight, catalyzed by the presence of copper. This metal could be present because it is in the hemocyanin of the blood of squids, the main food of sperm whales [1]. This oxidative degradation process generates highly appreciated products in perfumery known as ambroxides, substances that are usually obtained by distillation and that usually represent less than 1% by weight of the raw material. These include ambrafuran (**3**), also called ambroxide (ambrox^TM^), and other related compounds such as ambraoxide (**4**), ambracetal (**5**), and *cis*-α-ambrinol (**6**) (Figure 1b) [4,5]. Solid phase micro extraction (SPME) and gas chromatography-mass spectrometry (GC-MS) have allowed the identification of some of the more volatile natural components of ambergris, such as pristane, a fully saturated acyclic odorless nor-diterpene and γ-dihydroionone (**7**) [6], which has a similar structure to ambraaldehyde (**8**) (Figure 1b) [7].

The stereochemistry of ambrein (**2**) has been well established by chemical correlation methods [8]. In addition, the configuration of the tri- and tetracyclic natural products (**3**–**5**) derives from the bicyclic part or ambrein (**2**) (in red in Figure 1), while bi- and monocycles (**6**–**8**) derive from the left-hand-side monocyclic part of ambrein (in blue) [9]. *cis-*α-Ambrinol (**6**) has long attracted the attention of synthetic chemists due to its odor, which has been related to damp earth with a crude civet subnote [9]. It can be easily prepared by acid-catalyzed cyclization of γ-ionone, although in the process, β-ambrinol (**9**) and other cyclization products (**10**) are formed (Figure 2a) [10].

Enantiopure α-ambrinol has also been prepared using different approaches. In fact, both enantiomers of *cis*-α-ambrinol can be prepared by chemical resolution of the racemate through fractional crystallization of the camphanic derivatives (**11a**, **11b**) (Figure 2b) [11]. Both the natural (−)-(*S,S*)-*cis*-α-ambrinol (**(−)-6**) and the unnatural (+)-(*R,R*) enantiomer (**(+)-6**) have intense odor properties, although they are very different from each other [11]. In addition, (−)-*cis*-α-ambrinol (**(−)-6**) has also be prepared by acid-catalyzed cyclization of (+)-(*S*)-γ-dihydroionone (**(+)-7**) [10] (Figure 2c). This enantiopure compound is another highly appreciated component of ambergris that can also be obtained by enzymatic resolution of a derivative of commercial racemic α-ionone [10]. More recently, a hydrogen-bond assisted Brønsted-acid biocatalysis in water of neryl acetone (**13**) has allowed the enantiopure preparation of the unnatural enantiomer, (−)-(*R*)-γ-dihydroionone (**(−)-7**), which was cyclized into unnatural (+)-α-ambrinol (**(+)-6**) (Figure 2d) [12].

A different strategy, starting from commercially available geranylacetone (**14**), allowed the formation of γ-dihydroionone (**7**) through a Ti(III)-catalyzed radical cyclization (Figure 3) [13]. The enantioselective process is based on the preparation of chiral epoxide (**15**) through a Jacobsen’s asymmetric epoxidation followed by a Cp_2_TiCl-catalyzed stereoselective cyclization, a process that furnishes (**16**) in a highly stereoselective way. Cp_2_TiCl is a single electron transfer reagent that promotes a radical epoxide opening of (**15**) which proceeds with the retention of the configuration at the epoxide chiral center, leading to the secondary alcohol (**16**). Enantiomeric enrichment through kinetic resolution furnished the acetyl derivative (**17**). Its deoxygenation led to the protected (+)-γ-dihydroionone (**18**). Acid-catalyzed deprotection with concomitant cyclization yielded the desired (−)-*cis*-α-ambrinol **(−)-6**) [13].

## 2. Results and Discussion

As part of our continued effort in the preparation of marine natural products [14,15,16], and due to the fact that all the previously reported synthesis of *cis*-α-ambrinol (**6**) rely on an acid cyclization step that affords a mixture of regioisomers which is not easily purified by conventional chromatographic technics, we were interested in developing a concise diastereoselective total synthesis of *cis*-α-ambrinol (**6**).

The synthesis of compound **6** was planned according to the retrosynthesis depicted in Figure 4 through two alternative pathways, both of them using commercially available α-ionone (**19**) as a starting material. The first approach has as a key step a diastereoselective cyclization of the allylic carbonate (**20**) using the bimetallic system Cp_2_Ti^III^Cl/Pd^0^ (Figure 4a). The second one (Figure 4b), is based on a Barbier-type CpTi^III^Cl_2_-catalyzed intramolecular allylation of the chlorinated derivative (**22**).

The synthetic route based on the bimetallic system Ti(III)/Pd(0) is depicted in Figure 5. The first step is the regioselective reduction of the conjugated double in α-ionone (**19**), following a modification of a previously described methodology [17], to give α-dihydroinone (**12**). Epoxidation of (**12**) with *m*-CPBA afforded the epoxide (**21**) as a diastereoselective mixture *cis*:*trans* (86:12) in an 98% yield. Acid treatment of the *cis*-epoxide (***cis*-21**) gave 1-hydroxy-γ-dihydroionone (**23**) in a 62% yield. The formation of carbonate (**20**) was carried out using ethyl chloroformate under basic conditions, yielding the desired compound (**20**) in an 80% yield. The key step of this synthetic approach relies on a cooperative catalytic method [18,19]. We first tried the reaction using the combination Cp_2_TiCl/Ni(PPh_3_)_2_Cl_2_, a bimetallic system which has been described as an efficient promoter for the allylation of carbonyl compounds [19], although with little success in our case, as the process proved to be unproductive. However, treatment of allylic carbonate (**20**) with a source of Ti(III)/Pd(0) [18] (see Section 3 for details) gave a mixture of the desired natural *cis*-α-ambrinol (**6**) (37% yield), *trans*-α-ambrinol (**24**) (14% yield), and the monocyclic compound (**25**) (9% yield).

The relative stereochemistry of both isomers *cis*-α-ambrinol (**6**) and *trans*-α-ambrinol (**24**) was determined with the aid of NOE experiments (Figure 1). Especially relevant is the presence of correlations in (**6**) between the equatorial CH_3_-13 (δ = 1.24 ppm) and all four hydrogens in CH_2_-7 and CH_2_-9 while in (**24**), the observed correlations between the axial methyl CH_3_-13 (δ = 1.14 ppm) are only those with the β H (equatorial) of CH_2_-7 and CH_2_-9 in each case.

Although the yield of (**6**) was not completely satisfactory, the C-C bond formation reaction proceeds with high diastereoselectivity, and with simultaneous formation of the desired trisubstituted double bond. Spectroscopic data for synthetic *cis*-α-ambrinol (***cis*-6**) were identical to those of the natural compound [20,21,22].

The experimental results obtained in the cyclization of (**20**) with the Ti(III)/Pd(0) bimetallic system can be fully explained by the mechanism tentatively proposed in Figure 6. Initially, an oxidative addition of the allylic electrophile carbonate (**20**) to Pd^0^ would give the corresponding ƞ^3^-allyl-palladium intermediate **I**. Monoelectronic reduction of intermediate **I** by Cp_2_TiCl generates a ƞ^3^-allyl-palladium(II) intermediate, which could give the carbon-centered radical intermediate **II**, while the Pd^0^ complex is regenerated. This radical intermediate **II** could be trapped by a second molecule of Cp_2_TiCl to form two alkyl-Ti^IV^ species in metallotropic equilibrium (**III** and **IV**). β-Elimination of hydrogen in species **III** would account for the formation of monocyclic diene (**25**) and Cp_2_TiCl(H). It has been previously reported that this titanium hydride spontaneously decomposes to regenerate Cp_2_TiCl and molecular hydrogen [23]. On the other hand, the least sterically hindered alkyl-Ti^IV^ intermediate **IV** can evolve through two different rotational conformers, **IVa** and **IVb**. In one of them, **IVa**, the carbonyl group oxygen is arranged spatially close to the titanium atom (axial-like orientation), and therefore the intramolecular nucleophilic addition of the alkyl-Ti^IV^ to the ketone favors the diastereoselective formation of *cis*-α-ambrinol (**6**) as the main product. However, if the ketone oxygen is located at a greater distance from the titanium atom (equatorial-like orientation, **IVb**), the interaction between both atoms should be slightly weaker, resulting in a slower nucleophilic addition of the alkyl-Ti^IV^ intermediate (**IVb**) to the carbonyl group, and further leading to the formation of the diastereoisomer *trans*-α-ambrinol (**24**) as a minor product.

The synthetic advantage of this synthetic route is that it could lead to the enantiopure **(−)-6** using an enantiopure alcohol (−)-**23**. This was previously prepared by Serra [10] using a lipase-mediated racemic resolution.

Our second synthetic approach is based on a Barbier-type Ti(III)-catalyzed intramolecular allylation of an allyl chloride, which is summarized in Figure 7. In this case, we used a starting material α-dihydroionone (**12**), previously prepared in the other route.

Chlorination of **12** with NaClO afforded a diastereomeric mixture of allylic chlorides (**22**) in a 96% yield. With this substrate in hand, we first tried to induce the intramolecular allylation using an excess of Zn dust, which is known to react with allyl chlorides to form organometallic systems which can react with carbonyl groups. However, the reaction led to the formation of the bicyclic product (**26**), which originated as a result of the formation of a C-C bond between C1 and the carbonyl (C9). We next tried the allylation with two Ti(III) systems, the well-established single electron transfer reagent Cp_2_TiCl, and the half-sandwich titanocene CpTiCl_2_, both prepared by reduction with Mn of the appropriate Ti(IV) species.

The allylation was tested under catalytic and stoichiometric conditions for both systems. The results, summarized in Table 1, show a similar behavior in all cases, although CpTiCl_3_ seems to be superior both in terms of global yields and diastereoselectivity, particularly under stoichiometric conditions (Table 1, entry 3). In the light of these results, we decided to check whether the Ti(III)/Pd(0) combination strategy could be performed with the half-sandwich titanocene reagent CpTiCl_2_ using the ethylcarbonate (**20**) as a substrate. Indeed, the reaction proved to be successful (Table 1, entry 5), leading to a 73% global yield of the cyclic product, and with a 73:27 diastereoselectivity ratio using stoichiometric amounts of the Ti(III) source and catalytic of the Pd(0).

The diastereoselectivity observed in the cyclization of (**22**) to give (**6**) and (**24**) mediated or catalyzed by CpTiCl_2_ can be easily explained by a mechanism similar to the one discussed in Figure 6, although in this case the radical intermediate **II** would be formed by the homolytic cleavage of the activated C-Cl bond present in (**22**).

In conclusion, we have proved that *cis*-α-ambrinol (**6**) can be prepared from commercial α-ionone (**19**) with an overall yield of 46% in only three steps using a stoichiometric amount of CpTiCl_2_ for the intramolecular Barbier-type allylation of the chloro-derivative (**22**). *cis*-α-Ambrinol (**6**) can also be prepared in five steps from the same starting material through a Ti(III)/Pd(0) cyclization of the carbonate intermediate (**20**) with a 35% global yield. Finally, it should be mentioned that this synthesis of α-ambrinol (**6**) constitutes a new application of the usefulness of CpTiCl_2_ as a new monoelectronic transfer reagent, as we [24,25,26] and others [27,28] have previously reported.

## 3. Materials and Methods

### 3.1. General Details

THF was distilled from Na/benzophenone under argon, and in all experiments involving titanocene (III) was deoxygenated prior to use, and oven-dried glassware was used in all cases. NMR spectra were recorded on Bruker Nanobay Avance III HD 300 MHz, and Avance III HD 600 MHz spectrometers. Proton-decoupled ^13^C{^1^H} NMR and DEPT-135 were measured in all cases. When required, NOE 1D, COSY, HSQC and HMBC experiments were used for signal assignation. Chemical shifts (δ) are expressed in ppm and coupling constants (*J*) in hertzs (Hz). Chemical shifts are reported using CDCl_3_ as internal reference. IR Spectra were recorded with a Bruker Alpha spectrometer. Mass spectra were recorded in a Waters Xevo by LC-QTof-MS by electrospray ionization. All reactions were monitored by thin-layer chromatography (TLC) carried out on 0.2 mm DC-Fertigfolien Alugram^®^ XtraSil G/UV254 silica gel plates. The TLC plates were visualized with UV light and 7% phosphomolybdic acid or KMnO_4_ in water/heat. Flash chromatography was performed on silicagel 60 (0.04–0.06 mm). Hard copies of NMR and IR spectra can be found as Appendix A.

### 3.2. Synthesis of α-Dihydroionone *(**12**)*

To a solution of α-ionone (**19**) (864 mg, 4.5 mmol) in THF (10 mL) was added Ni-Raney (0.4 g). The mixture was stirred under H_2_ (1 atm) for 30 min at room temperature in a hydrogenation apparatus. The mixture was filtered through celite, and the solvent evaporated, yielding (**12**) (873 mg, 4.5 mmol, 100%) as a colorless oil. Spectroscopic data are in agreement with literature values [29].

IR (ATR) *v* (cm^−1^): 2954, 2915, 2870, 1714, 1449, 1361, 1252, 1218, 1159, 961, 944, 811, 555. 

^1^H NMR (300 MHz, CDCl_3_) δ (ppm): 5.36 (1H, bs, H1), 2.50 (1H, dd, *J* = 3.1, 7.0 Hz H8a), 2.47 (1H, dd, *J* = 2.0, 6.7 Hz H8b), 2.16 (3H, s, H9), 1.98 (2H, m), 1.85–1.73 (1H, m), 1.69 (3H, q, *J* = 1.7 Hz H13), 1.67–1.59 (1H, m), 1.51–1.48 (1H, m), 1.45–1.38 (1H, m), 1.19–1.12 (1H, m), 0.94 (3H, s), 0.89 (3H, s).

^13^C NMR (75 MHz, CDCl_3_) δ (ppm): 209.2 (C, C9), 135.6 (C, C6), 121.1 (CH, C1), 48.5 (CH), 43.8 (CH_2_), 32.6 (C, C4), 31.5 (CH_2_), 30.0 (CH_3_), 27.7 (CH_3_), 27.6 (CH_3_), 24.4 (CH_2_), 23.5 (CH_3_), 23.0 (CH_2_).

### 3.3. Preparation of Epoxide ***21***

To a solution of α-dihydroionone (12) (3.05 g, 15.67 mmol) in anhydrous CH_2_Cl_2_ (60 mL) at 0 °C, MCPBA (4.25 g, 17.24 mmol) was added. The mixture was stirred under N_2_ and allowed to reach room temperature for 3 h. The reaction was quenched by stirring for 15 min. with saturated NaHCO_3_ (30 mL) and another 30 mL of Na_2_S_2_O_3_ (10% in water). The two phases were separated and the organic layer was washed with brine, dried over anhydrous MgSO_4_ and the solvent was removed in a vacuum to give (**21**) as a mixture 86:12 *cis:trans* (3.21 g, 98%). Compound (***cis****-***21**) was purified by column chromatography (hexane:EtOAc 9:1) (83% yield from (**12**)). Colorless oil. Spectroscopic data are in agreement with literature values [17].

IR (ATR) *v* (cm^−1^): 2962, 2932, 2871, 1713, 1449, 1363, 1233, 1181, 1160, 1097, 1040, 995, 899, 864, 747, 557, 526.

^1^H NMR (600 MHz, CDCl_3_) δ (ppm): 2.96 (1H, s, H1), 2.76 (1H, ddd, *J* = 16.2, 10.1, 5.3 Hz, H8a), 2.52 (1H, ddd, *J* = 16.2, 9.9, 5.8 Hz, H8b), 2.18 (3H, s, H10), 1.94 (1H, dd, *J* = 15.5, 6.0 Hz, H2a), 1,87–1.81 (1H, m, H2b), 1.75–1.70 (1H, m, H7a), 1.61–1.54 (1H, m, H7b), 1.40 (1H, dd, *J* = 10.7, 6.0 Hz, H5), 1.34 (3H, s, H13), 1.32–1.27 (1H, m, H3a), 0.90 (3H, s, H11), 0.85 (3H, s, H12), 0.83 (1H, m, H3b).

^13^C NMR (75 MHz, CDCl_3_) δ (ppm): 209.4 (C, C9), 60.1 (CH, C1), 59.3 (C, C6), 46.2 (CH, C5), 43.1 (CH_2_, C8), 31.6 (C, C4), 30.1 (CH_3_, C10), 27.9 (CH_3_, C12), 27.3 (CH_3_, C11), 27.2 (CH_3_, C13), 26.7 (CH_2_, C3), 22.1 (CH_2_, C2), 21.6 (CH_2_, C7).

### 3.4. Synthesis of 1-Hydroxy-γ-dihydroionone *(**23**)*

To a solution of compound (***cis*-21**) (934 mg, 4.44 mmol) in CH_2_Cl_2_ (30 mL) at 0 °C, *p*-TSA (76 mg, 0.44 mmol) was added. The mixture was stirred for 12 h (0 °C), and then *p*-TSA (76 mg, 0.44 mmol) was again added and stirred for 4 h at room temperature. The mixture was washed with saturated NaHCO_3_ (10 mL × 3) and dried over anhydrous MgSO_4_. The solvent was removed in a vacuum and the residue purified by silica gel flash column chromatography (hexane/EtOAc, 8:2) to afford alcohol (**23**) (575 mg, 62%) as a colorless oil. Spectroscopic data are in agreement with literature values [30].

IR (ATR) *v* (cm^−1^): 3412, 2934, 2867, 1707, 1648, 1456, 1410, 1363, 1159, 1062, 1038, 898, 639, 584, 502. 

^1^H NMR (600 MHz, CDCl_3_) δ (ppm): 5.18 (1H, s, H13a), 4.65 (1H, s, H13b), 3.96 (1H, dd, *J =* 4.8, 9.0 Hz, H1), 2.58 (1H, ddd, *J =* 5.4, 9.0, 18.0 Hz, H8a), 2.34 (1H, ddd, *J =* 7.8, 8.4, 18.0 Hz, H8b), 2.11 (3H, s, H10), 1.91 (1H, m, H2a), 1.85 (1H, m, H7a), 1.74–1.62 (3H, m, H7b, H5, OH), 1.50 (1H, m, H3a), 1.46 (1H, m, H2b), 1.38 (1H, m, 3b), 0.98 (3H, s, H11), 0.74 (3H, s, H12).

^13^C NMR (75 MHz, CDCl_3_) δ (ppm): 209.5 (C, C9), 150.3 (C, C6), 105.5 (CH_2_, C13), 73.6 (CH, C1), 51.3 (CH, C5), 42.5 (CH_2_, C8), 37.9 (CH_2_, C3), 35.8 (C, C4), 33.2 (CH_2_, C2), 30.0 (CH_3_, C10), 29.3 (CH_3_, C11), 21.0 (CH_3_, C12), 19.5 (CH_2_, C7).

### 3.5. Synthesis of Ethyl Carbonate *(**20**)*

In an N_2_ atmosphere at 0 °C, ethyl chloroformate (0.74 mL, 7.59 mmol), pyridine (1.54 mL, 18.98) and DMAP (64 mg. 0.51 mmol) were added to a solution of (**23**) (532 mg, 2.53 mmol) in CH_2_Cl_2_ (40 mL). After 10 min, the cooling bath was removed, and the mixture was stirred for 20 h. TIt was then diluted with Et_2_O and washed with HCl (3%) and water. The organic layer was dried over anhydrous MgSO_4_, the solvent was removed in a vacuum and the residue purified by silica gel flash column chromatography (gradient hexane/EtOAc) to afford carbonate (**20**) (571 mg, 80%) as a colorless oil.

IR (ATR) *v* (cm^−1^): 2949, 2871, 1741, 1715, 1650, 1456, 1367, 1251, 1162, 1007, 903, 856, 790. HREIMS (*m*/*z*) calcd. for C_16_H_26_O_4_ 282.1831 [M]^+^, found 282.1833.

^1^H NMR (300 MHz, CDCl_3_) δ (ppm): 5.17 (1H, s, H13a), 4.96 (1H, dd, *J =* 4.5, 8.2 Hz, H1), 4.73 (1H, s, H13b), 4.22 (1H, q, *J =* 7.1 Hz, OCH_2_), 2.54 (1H, ddd, *J =* 4.5, 9.7, 17.6 Hz, H8a), 2.31 (1H, m, H8b), 2.13 (3H, s, H10), 1.98–1.83 (2H, m, H2a, H7a), 1.75–1.57 (4H, m, H2b, H7b, H5, H3a), 1.41 (1H, dd, *J* = 4.5, 10.3 Hz, H3b), 1.34 (3H, t, *J* = 7.1 Hz), 1.00 (3H, s, H11), 0.82 (3H, s, H12).

^13^C NMR (75 MHz, CDCl_3_) δ (ppm): 209.2 (C, C9), 154.5 (C, OCOO), 144.8 (C, C6), 109.2 (CH_2_, C13), 78.5 (CH, C1), 63.8 (CH_2_, OCH_2_), 51.4 (CH, C5), 42.3 (CH_2_, C8), 35.3 (C, C4), 30.0 (CH_3_, C10), 29.4 (CH_2_), 28.8 (CH_3_), 22.6 (CH_3_), 20.0 (CH_2_), 14.3 (CH_3_, OCH_2_CH_3_).

### 3.6. Synthesis of Ambrinol Using Cp_2_TiCl_2_

Deoxygenated, dry THF (6.5 mL) was added to a mixture of Cp_2_TiCl_2_ (195 mg, 0.76 mmol), Pd(PPh_3_)_2_Cl_2_ (54 mg, 0.076 mmol), and Mn dust (169 mg, 3.04 mmol) in an Ar atmosphere and the red suspension was stirred at room temperature until it turned lime green (after about 15 min). Then, a solution of (**20**) (107 mg, 0.38 mmol) in THF (2 mL) was added dropwise and the mixture was stirred for four days. The reaction was diluted in EtOAc, washed with HCl 12% and brine, dried (anhydrous Na_2_SO_4_) and the solvent was removed. The residue was purified by flash chromatography (hexane/EtOAc 9:1) yielding *cis*-α-ambrinol (**6**) (27 mg, 37%), *trans-*α-ambrinol (**24**) (10 mg, 14%) and compound (**25**) (7 mg, 9%).

*cis*-α-ambrinol (**6**): Colorless oil; spectroscopic data are in agreement with literature values [12]. IR (ATR) *v* (cm^−1^): 3449, 2957, 2913, 2868, 2843, 1451, 1383, 1363, 1321, 1263, 1237, 1186, 1134, 1111, 1097, 1019, 995, 926, 912, 878, 800, 727, 501. HREIMS (*m*/*z*) calcd. for C_13_H_22_O 194.1671 [M]^+^, found 194.1670.

^1^H NMR (600 MHz, CDCl_3_) δ (ppm): 5.48 (1H, s, H1), 2.15 (1H, bd, *J* = 13.2 Hz, H9a), 2.10 (1H, dd, *J* = 13.2, 2.5 Hz, H9b), 2.04–2.00 (2H, m, H2), 1.87 (1H, s, OH), 1.76–1.69 (2H, m, H6a, H7eq), 1.52 (1H, bd, *J* = 12.6 Hz, H5), 1.47 (1H, td, *J* = 13.4, 4.2 Hz, H7ax), 1.39 (1H, dt, *J* = 13.0, 7.3 Hz, H3eq), 1.29 (1H, td, *J* = 13.0, 3.4 Hz, H6b), 1.24 (3H, s, H13), 1.21 (1H, m, H3ax), 0.94 (3H, s, H11), 0.89 (3H, s, H12). 

^13^C NMR (151 MHz, CDCl_3_) δ (ppm): 137.5 (C, C10), 122.2 (CH_2_, C1), 70.4 (C, C8), 50.0 (CH_2_, C9), 47.4 (CH, C5), 39.1 (CH_2_, C7), 33.4 (CH_2_, C3), 31.2 (C, C4), 29.4 (CH_3_, C13), 28.2 (CH_3_, C11), 26.1 (CH_3_, C12), 25.2 (CH_2_, C6), 22.9 (CH_2_, C2).

*trans*-α-ambrinol (**24**): Colorless oil; IR (ATR) *v* (cm^−1^): 3368, 2923, 2869, 1710, 1664, 1451, 1364, 1260, 1198, 1119, 1062, 1020, 1010, 947, 919, 831, 820, 727, 682, 571. HREIMS (*m*/*z*) calcd. for C_13_H_22_O 194.1671 [M]^+^, found 194.167.

^1^H NMR (600 MHz, CDCl_3_) δ (ppm): 5.37 (1H, s, H1), 2.21 (1H, dd, *J* = 12.5, 2.6 Hz, H9eq), 2.13 (1H, bd, *J* = 12.5 Hz, H9ax), 1.97 (2H, m, H2), 1.81–1.75 (2H, m, H7eq, H6eq), 1.60 (1H, bd, *J* = 13.2 Hz, H5), 1.55 (1H, td, *J* = 13.3, 4.3 Hz, H7ax), 1.34 (1H, dt, *J* = 13.0, 6.1 Hz, H3eq), 1.25 (2H, m, H3ax. OH), 1.14 (3H, s, H13), 1.10 (1H, td, *J* = 6.5, 1.9 Hz, H6eq), 0.94 (3H, s, H11), 0.84 (3H, s, H12). 

^13^C NMR (151 MHz, CDCl_3_) δ (ppm): 137.4 (C, C10), 120.9 (CH_2_, C1), 72.1 (C, C8), 50.9 (CH_2_, C9), 47.0 (CH, C5), 40.8 (CH_2_, C7), 35.2 (CH_2_, C3), 31.3 (C, C4), 28.7 (CH_3_, C11), 25.6 (CH_3_, C13), 25.3 (CH_2_, C6), 24.6 (CH_3_, C12), 22.8 (CH_2_, C2).

Compound (**25**): Colorless oil [31]; ^1^H NMR (300 MHz, CDCl_3_) δ 6.01 (1H, m, H1), 5.67 (1H, m, H2), 4.91 (1H, s, H13a), 4.70 (1H, s, H13b), 2.55–2.32 (2H, m), 2.14 (3H, s, H10), 1.84–1.70 (3H, m), 1.35–1.25 (2H, m), 1.02 (3H, s, H11), 0.88 (3H, s, H12). 

### 3.7. Synthesis of Ambrinol Using CpTiCl_3_

Deoxygenated, dry THF (6 mL) was added to a mixture of CpTiCl_3_ (137 mg, 0.62 mmol), Pd(PPh_3_)_2_Cl_2_ (44 mg, 0.062 mmol), and Mn dust (138 mg, 2.48 mmol) in an Ar atmosphere red suspension was stirred at room temperature until it turned lime green (after about 15 min). Then, a solution of (**20**) (88 mg, 0.31 mmol) in THF (0.5 mL) was added dropwise and the mixture was stirred for 19 h. The reaction was quenched and purified as described above for Cp_2_TiCl_2_. Compound (**6**) (32 mg, 0.16 mmol, 53%) and compound (**24**) (12 mg, 0.06 mmol, 20%) were obtained.

### 3.8. Synthesis of 1-Chloro-γ-dihydroionone *(**22**)*

To a solution of α-dihydroionone (**12**) (455 mg, 2.32 mmol) in hexane (1.5 mL), NaClO (aqueous solution 15%) (3 mL, 12.9 mmol) was added. The mixture was cooled at 0 °C and H_3_PO_4_ (aqueous solution 40%) (0.4 mL, 1.72 mmol) was added. After stirring for 2 h at 0 °C, water (15 mL) was added, and the mixture extracted with Et_2_O (15 mL × 3). The combined organic layer was washed with brine and dried over anhydrous MgSO_4_. The solvent was removed in a vacuum to give 1-chloro-γ-dihydroionone (**22**) as a mixture of *cis* (*R*S**): *trans* (*R*R**) isomers (10:0.8 ratio); colourless oil (0.51 g, 96%). Spectroscopic data are in agreement with literature values [32].

IR (ATR) *v* (cm^−1^): 2950, 2869, 1714, 1649, 1454, 1419, 1388, 1364, 1234, 1158, 989, 909, 886, 760, 705, 602, 537. 

^1^H NMR (300 MHz, CDCl_3_) δ (ppm) (*R*S**) isomer signals: 5.34 (1H, s, H13a), 4.78 (1H, s, H13b), 4.48 (1H, dd, *J* = 4.8, 6.6 Hz, H1), 0.94 (3H, s), 0.86 (3H, s); (*R*R**) isomer signals: 5.22 (1H, s, H13a), 4.72 (1H, s, H13b), 4.62 (1H, t, *J* = 5.0 Hz, H1), 1.01 (3H, s), 0.79 (3H, s); both isomer signals: 2.64–2.53 (1H, m), 2.38–2.27 (1H, m), 2.16–2.06 (1H, m), 2.12 (3H, s, H10), 1.92–1.69 (5H, m), 1.30 (1H, ddd, *J =* 4.3, 8.3, 13.0 Hz).

^13^C NMR (75 MHz, CDCl_3_) δ (ppm): 209.3 (C, C9), 145.8 (C, C6), 111.2 (CH_2_, C13), 63.0 (CH, C1), 52.2 (CH, C5), 42.2 (CH_2_), 35.2 (CH_2_), 35.0 (C, C1), 33.9 (CH_2_), 30.1 (CH_3_), 28.5 (CH_3_), 21.3 (CH_2_).

### 3.9. Zinc Cyclization of 1-Chloro-γ-dihydroionone *(**22**)*

Zn (161 mg, 2.4 mmol) and HCOONa (169 mg, 2.4 mmol) were added to a solution of (**22**) (138 mg, 0.6 mmol) in THF (1 mL) and EtOH (3 mL). The mixture was heated at 80 °C and stirred overnight. The reaction was cooled, diluted with Et_2_O (10 mL) and filtered through celite. The celite was then washed with H_2_O (10 mL), the two phases were separated, and the aqueous layer was extracted with Et_2_O (10 mL × 3). The combined organic layer was washed with saturated NaHCO_3_ (30 mL) and brine (30 mL), dried over anhydrous MgSO_4_ and the solvent was removed in a vacuum. The residue was purified by flash chromatography (hexane/Et_2_O 9:1) afforded compound (**26**) (30 mg, 26%) as a colorless oil.

IR (ATR) *v* (cm^−1^): 3473, 3065, 2969, 2932, 2867, 1717, 1653, 1479, 1452, 1384, 1364, 1246, 1211, 1168, 1106, 983, 916, 882, 693, 541. HREIMS (*m*/*z*) calcd. for C_13_H_22_O 194.1671 [M]^+^, found 194.1675.

^1^H NMR (300 MHz, CDCl_3_) δ (ppm): 4.84 (2H, m), 2.12 (1H, m), 2.00–1.93 (1H, m), 1.89 (2H, m), 1.81–1.75 (2H, m), 1.73–1.64 (2H, m), 1.54 (1H, dd, *J =* 14.0, 7.1 Hz), 1.25 (3H, s), 1.21–1.19 (1H, m), 1.00 (3H, s), 0.93 (3H, s).

^13^C NMR (75 MHz, CDCl_3_) δ (ppm): 152.1 (C), 108.5 (CH_2_), 73.8 (C), 51.2 (CH), 49.3 (CH), 35.3 (C), 35.3 (CH_2_), 33.9 (CH_2_), 29.0 (CH_3_), 28.9 (CH_3_), 26.8 (CH_3_), 26.0 (CH_2_), 25.1 (CH_2_).

### 3.10. CpTiCl_3_-Catalyzed Cyclization of 1-Chloro-γ-dihydroionone *(**22**)*

Deoxygenated, dry THF (3 mL) was added to a mixture of CpTiCl_3_ (11 mg, 0.052 mmol) and Mn dust (57 mg, 1.04 mmol) in an Ar atmosphere resulting in a green suspension. Me_3_SiBr (0.22 mL, 0.52 mmol) was then added, and the mixture turned turquoise. Subsequently, a solution of (**22**) (119 mg, 0.52 mmol) in THF (1 mL) was added dropwise and the mixture was stirred for 4 h. The reaction was filtered, diluted in Et_2_O, washed with HCl 3% and brine, dried (anhydrous MgSO_4_), and the solvent was removed. The residue was purified by flash chromatography (hexane/EtOAc 8:2) afforded *cis*-α-ambrinol (**6**) (42 mg, 42%) and *trans*-α-ambrinol (**24**) (18 mg, 18%).

### 3.11. CpTiCl_3_-Induced Cyclization of 1-Chloro-γ-dihydroionone *(**22**)*

Deoxygenated, dry THF (3 mL) was added to a mixture of CpTiCl_3_ (105 mg, 0.48 mmol) and Mn dust (53 mg, 0.96 mmol) in an Ar atmosphere resulting in a green suspension. Subsequently, a solution of (**22**) (110 mg, 0.48 mmol) in THF (1 mL) was added dropwise and the mixture was stirred for 5 h. The reaction was quenched and purified as indicated for the catalytic version to give *cis*-α-ambrinol (**6**) (45 mg, 48%) and *trans*-α-ambrinol (**24**) (18 mg, 19%). 

### 3.12. Cp_2_TiCl_2_ Cyclization of 1-Chloro-γ-dihydroionone *(**22**)*

For the catalytic cyclization, deoxygenated and dry THF (3 mL) was added to a mixture of Cp_2_TiCl_2_ (25 mg, 0.093 mmol) and Mn dust (160 mg, 2.85 mmol) in an Ar atmosphere and the red suspension was stirred at room temperature until it turned lime green (after about 15 min). Then, a mixture of 2,4,6-collidine (0.36 mL, 2.69 mmol) and Me_3_SiCl (0.2 mL, 1.54 mmol) in THF (1 mL) was added. Subsequently, a solution of (**22**) (110 mg, 0.48 mmol) in THF (1 mL) was added dropwise and the mixture was stirred for 1 h 45 min. The reaction was filtered, diluted in Et_2_O, washed with HCl 3% and brine, dried (anhydrous MgSO_4_), and the solvent was removed. The residue was purified by flash chromatography (hexane/EtOAc 8:2) affording *cis*-α-ambrinol (**6**) (31 mg, 33%) and *trans*-α-ambrinol (**24**) (16 mg, 17%).

With stoichiometric amounts of Cp_2_TiCl_2_, the reaction was carried out under the same conditions as above, employing Cp_2_TiCl_2_ (247 mg, 0.93 mmol), Mn dust (162 mg, 2.85 mmol) and (**22**) (100 mg, 0.44 mmol). The mixture was stirred for 2 h, quenched and purified as indicated above, and *cis*-α-ambrinol (**6**) (32 mg, 37%) and *trans*-α-ambrinol (**24**) (15 mg, 18%) were obtained.

## 4. Conclusions

Natural *cis*-α-ambrinol (**6**) can be diastereoselectively prepared by the intramolecular Barbier-type reaction of a carbonate derivative of 1-hydroxy-γ-dihydroionone (**23**) promoted by the bimetallic system based on Ti(III)/Pd(0) [Cp_2_TiCl, Pd(PPh_3_)_2_Cl_2_]. Although the yield is somewhat low, the process can lead to enantiopure *cis*-α-ambrinol (**6**) if compound (**23**) is resolved enzymatically. In addition, CpTiCl_2_-promoted cyclization of 1-chloro-γ-dihydroionone (**22**) leads to the formation of α-ambrinol in a 64% global yield as a separable mixture of *cis* and *trans* diastereomers in a 2.3:1 ratio, which accounts for a 46% global yield from commercial α-ionone for the natural diastereoisomer *cis*-α-ambrinol (**6**).

## Data Availability

Data is contained within the article and Appendix A.

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
