# Peer review of "A Concise Diastereoselective Total Synthesis of α-Ambrinol"

_marinedrugs, 2023, doi:10.3390/md21040230_

Round 1
Reviewer 1 Report
This manuscript describes a concise diastereoselective total synthesis of α-ambrinol via an intramolecular Barbier-type allylation of chlorinated derivatives using stoichiometric CpTiCl2. Although the desired cis-α-ambrinol can be prepared from α-ionone in only three steps in this synthesis, the diastereoselectivity in the final step was not good. In addition, this synthesis obtained a reacemate of cis-α-ambrinol rather than the enantiomerically enriched cis-α-ambrinol. Therefore, I do not recommend the publication of this manuscript in Marine Drugs.
Author Response
We wish to express our gratitude to the reviewer for the effort of considering our manuscript. We agree that the method provides a racemate. We have clearly stated in the introduction that there are already described enantioselective methods for the preparation of α-ambrinol. And also, that one of our intermediates (alcohol 23) has been already prepared as a single enantiomer by the group of Serra et al. The combination of the literature method for the preparation of this intermediate with the synthesis described here constitutes a new “formal” enantioselective total synthesis of α-ambrinol, although we haven’t claimed it as such in the manuscript.
However, we have proved that both diastereoisomers can be separated by simple chromatography, and as far as we know, this is the first time that they have been fully characterized. This has allowed us to present a comparison of their spectroscopic properties, which is useful as they are sometimes confused in literature. In addition, the Ti(III) mediated cyclizations described in table 1 show a clear preference for the formation of the cis diastereoisomer, up to 73:27, which means that the reaction is diastereoselective at the level described in the manuscript.
Reviewer 2 Report
The manuscript of López-Martínez et al. presents a new approach towards total synthesis of α-ambrinol starting from commercially available α-ionone. The manuscript is written in an acceptable style so that it can be easily read and followed, and this especially applies to the experimental part and is not a problem to follow.
Reading the text I came across a few mistakes, which should be corrected.
Please, change the Z/E stereodescriptors on line 331, 335 and 336 to R/S: Z → R, E → S. As per IUPAC recommendations.
Complete characterization data must be given for new compounds in 3. Materials and Methods. Please, add HRMS and [a]D data.
Line 122, instead of 80%, it should be 62% (as it says in the lines 110 and 256). Also in the same line (i) should be deleted.
Line 315, instead of 7.70, it should be 1.70.
In 3. Materials and Methods, some compound numbers are not bolded.
In my opinion the manuscript can be accepted for publication with minor revision as suggested.
Author Response
We wish to express our gratitude to the reviewer for the obvious pains taken in the process of reading and exhaustive checking our manuscript. We have addressed the comments and suggestions as follows:
Reading the text I came across a few mistakes, which should be corrected.
- Please, change the Z/E stereodescriptors on line 331, 335 and 336 to R/S: Z → R, E → As per IUPAC recommendations.
- We have changed line 331, which now reads: as a mixture of cis (R*S*) : trans (R*R*) isomers (10:0.8 ratio);
- We have changed line 335-336, which now reads: (R*S*) isomer signals: .......; (R*R*) isomer signals:......
- Complete characterization data must be given for new compounds in 3. Materials and Methods. Please, add HRMS and [a]D
- We have added HRMS data for new compounds (20, 6, 24 and 26) in Materials and Methods section. There is no need to add [a]D as all prepared compounds are racemic.
- Line 122, instead of 80%, it should be 62% (as it says in the lines 110 and 256). Also in the same line (i) should be deleted.
- We have changed line 122, which now reads: (c) p-TSA, 62%;
- Line 315, instead of 7.70, it should be 1.70.
- We have changed line 315, which now reads: 1.84-1.70 (3H, m)
- In 3. Materials and Methods, some compound numbers are not bolded.
- The following compounds numbers have been bolded:
line 217 (12)
line 231 (21)
line 237 (21)
line 238 (cis-21)
line 239 (12)
line 250 (23)
line 267 (20)
line 325 (22)
line 343 (22)
line 360 (22)
line 369 (22)
line 376 (22)
Reviewer 3 Report
Authors described synthesis of racemic cis-alfa-ambrinol from alfa-ionone. The key step was Ti(III)/Pd(0) cyclization of the carbonate derivative or CpTiCl2 mediated intramolecular Barbier allylation. Though the yield in both cases is moderate, the synthesis is an interesting application of CpTiCl2. Additionally authors determined relative stereochemistry of cis/trans isomers of alfa-ambrinol using NOE experiments.
Comments:
1) Authors should remove unnecessary selfcitation. Overall 13 out of 35 citations are selfcitations.
2) Correct dihidroionone to dihydroionone - line 107, 166 and scheme 7.
3) Check line 122 c) (i) ?
4) Please add physical state and the color of the cmpds in Experimental.
5) Please add references for spectral data of known cmpds in the Experimental.
Author Response
We wish to express our gratitude to the reviewer for the obvious pains taken in the process of reading and exhaustive checking our manuscript. We have addressed the comments and suggestions as follows:
1) Authors should remove unnecessary selfcitation. Overall 13 out of 35 citations are selfcitations.
We have removed 7 references :15, 18, 19, 20, 21, 22 and 23 in line 92.
2) Correct dihidroionone to dihydroionone - line 107, 166 and scheme 7.
- We have corrected this typo in lines 107, 166 and scheme 7;
3) Check line 122 c) (i) ?
- We have changed line 122, which now reads: (c) p-TSA, 62%;
4) Please add physical state and the color of the cmpds in Experimental.
- We have added physical state and color to all compounds in Materials and Methods section;
5) Please add references for spectral data of known cmpds in the Experimental.
- We have added references for spectral data of known compounds (12, 21, 22, 23, 25 and 6) in Materials and Methods section. (new references 29, 17, 32, 30, 31 and 12 respectively.
Round 2
Reviewer 1 Report
In this revised manuscript, a number of suggestions and comments from former reviewers have been considered in this new manuscript. Therefore, this manuscript is recommended for publication after minor revisions.
In Scheme 6, the palladium reagent for the oxidative addition of 20 should be Pd(0) (not Pd(PPh3)2Cl2), and the intermediate I should be contained Pd(II) (not Pd(IV)). The single-electron reduction of intermediate I by Cp2TiCl will regenerate Pd(0) instead of Pd(PPh3)2Cl2.
The double bond is missing in the intermediates IVa and IVb in Scheme 6.
Radical intermediate II could also be formed from 22, please also draw this in Scheme 6.
In Scheme 7 and experimental section, “NaHCO2” should be revised to “HCOONa”
Author Response
We wish to express our gratitude to the reviewer for the effort of considering our manuscript. We have addressed the comments and suggestions as follows:
- In Scheme 6, the palladium reagent for the oxidative addition of 20 should be Pd(0) (not Pd(PPh3)2Cl2), and the intermediate I should be contained Pd(II) (not Pd(IV)). The single-electron reduction of intermediate I by Cp2TiCl will regenerate Pd(0) instead of Pd(PPh3)2Cl2.
Thank you very much for noticing this error. We are very sorry for it. We have now fixed the oxidation states of the species involved in Scheme 6.
- The double bond is missing in the intermediates IVa and IVb in Scheme 6.
Thank you very much for noticing this error. We have now drawn the double bonds required in Scheme 6.
- Radical intermediate II could also be formed from 22, please also draw this in Scheme 6.
Thank you very much for noticing this error. We have properly modified Scheme 6.
- In Scheme 7 and experimental section, “NaHCO2” should be revised to “HCOONa”.
Thank you very much for noticing this error. We have properly modified Scheme 7 legend and experimental section.